# Cytotoxicity of the effector protein BteA was attenuated in *Bordetella pertussis* by insertion of an alanine residue

Jan Bayram[1], Ivana Malcova[1], Larisa Sinkovec[1], Jana Holubova[1], Gaia Streparola[2], David Jurnecka[1], Jan Kucera[2], Radislav Sedlacek[2], Peter Sebo[1], Jana Kamanova[1]*

1 Institute of Microbiology of the Czech Academy of Sciences, Prague, Czech Republic, 2 Czech Centre for Phenogenomics, Institute of Molecular Genetics of the Czech Academy of Sciences, Vestec, Czech Republic

* kamanova@biomed.cas.cz

**Data Availability Statement:** All relevant data are within the manuscript and its Supporting Information files.

## Abstract

*Bordetella bronchiseptica* and *Bordetella pertussis* are closely related respiratory pathogens that evolved from a common bacterial ancestor. While *B. bronchiseptica* has an environmental reservoir and mostly establishes chronic infections in a broad range of mammals, *B. pertussis* is a human-specific pathogen causing acute pulmonary pertussis in infants and whooping cough illness in older humans. Both species employ a type III secretion system (T3SS) to inject a cytotoxic BteA effector protein into host cells. However, compared to the high BteA-mediated cytotoxicity of *B. bronchiseptica*, the cytotoxicity induced by *B. pertussis* BteA (*Bp* BteA) appears to be quite low and this has been attributed to the reduced T3SS gene expression in *B. pertussis*. We show that the presence of an alanine residue inserted at position 503 (A503) of *Bp* BteA accounts for its strongly attenuated cytotoxic potency. The deletion of A503 from *Bp* BteA greatly enhanced the cytotoxic activity of *B. pertussis* B1917 on mammalian HeLa cells and expression of *Bp* BteAΔA503 was highly toxic to *Saccharomyces cerevisiae* cells. *Vice versa*, insertion of A503 into *B. bronchiseptica* BteA (*Bb* BteA) strongly decreased its cytotoxicity to yeast and HeLa cells. Moreover, the production of *Bp* BteAΔA503 increased virulence of *B. pertussis* B1917 in the mouse model of intranasal infection (reduced LD50) but yielded less inflammatory pathology in infected mouse lungs at sublethal infectious doses. This suggests that A503 insertion in the T3SS effector *Bp* BteA may represent an evolutionary adaptation that fine-tunes *B. pertussis* virulence and host immune response.

## Author summary

Pertussis remains the least-controlled vaccine-preventable infectious disease and the mechanisms by which *Bordetella pertussis* subverts defense mechanisms of human airway mucosa remain poorly understood. We found that *B. pertussis* had the cytotoxic activity of its type III secretion system-delivered effector BteA strongly attenuated by insertion of an alanine residue at position 503 as compared to the BteA homologue of the animal

**Funding:** This work was supported by grants 18-16772Y to JKa and 19-27630X to PS of the Czech Science Foundation (www.gacr.cz) and projects LM2018133 (Czech National Node to the European Infrastructure for Translational Medicine) to PS, LM2015040 (Czech Centre for Phenogenomics) to RS and CZ.1.05/2.1.00/19.0395 ('Higher quality and capacity for transgenic models') to RS of the Ministry of Education, Youth and Sports of the Czech Republic, Sports (www.msmt.cz). The funders had no role in study design, data collection and analysis, decision to publish, or preparation of the manuscript.

**Competing interests:** The authors have declared that no competing interests exist.

pathogen *B. bronchiseptica*. This functional adaptation reduced the capacity of *B. pertussis* to suppress host inflammatory response and may contribute to an acute course of the pulmonary form of human infant pertussis.

## Introduction

The genus *Bordetella* embraces three closely related mammalian pathogens, the so-called classical bordetellae, *B. pertussis*, *B. parapertussis*, and *B. bronchiseptica* that cause respiratory infections of various symptoms, duration, and severity. The strictly human-adapted *B. pertussis* species is the causative agent of pertussis or whooping cough, a highly contagious respiratory illness that remains of primary concern for public health. Pertussis used to be the prime cause of infant mortality in the pre-vaccine era and despite high vaccination coverage, it remains one of the least controlled vaccine-preventable infectious diseases. Pertussis is currently resurging in the developed countries that switched from the use of whole-cell pertussis vaccine to the use of less reactogenic acellular pertussis vaccine [1, 2]. Older children and adults with waning vaccine-induced immunity develop a whooping cough disease, where the severe paroxysmal coughing can last for up to three months, occasionally yielding rib fracture or even death of the elderly [3]. Most deaths (~95%) to pertussis, however, occur in unvaccinated infants below 3 months of age, which typically develop a marked leukocytosis, bronchopneumonia, and refractory pulmonary hypertension eventually leading to cardiac failure [4]. The *B. parapertussis* species consists of two distinct lineages, those that infect sheep and those that cause a whooping cough illness in humans [5–7]. The *B. bronchiseptica* species infects a broad range of mammals, including humans, and consists of 2 distinct subpopulations, designed as a complex I and complex IV [8]. These bacteria elicit pathologies ranging from typical chronic and often asymptomatic respiratory infections, up to acute illnesses, such as kennel cough in dogs, or bronchopneumonia, and atrophic rhinitis in piglets [3, 9].

The three classical *Bordetella* species provide a unique opportunity to study the ongoing bacterial evolution, as *B. pertussis* and *B. parapertussis* appear to have evolved independently and rather recently from a *B. bronchiseptica*-like ancestor [8]. All three species share a number of virulence factors, such as adhesins (e.g. the filamentous hemagglutinin (FHA), pertactin, fimbriae, and other) and protein toxins, such as the adenylate cyclase toxin (CyaA) and the dermonecrotic toxin (DNT) [10]. However, some virulence factors are expressed by just one of these species, as exemplified by the pertussis toxin (PT) produced only by *B. pertussis*. Expression of some virulence factors is then differently regulated in different species, or even subpopulations of the same species, such as the genes coding for the type III secretion system (T3SS) [10–13].

The T3SS nanomachine functions as an injectisome that crosses the bacterial cell wall and allows for the delivery of bacterial effector proteins directly from bacterial cytosol into the host cell cytoplasm. The T3SS-delivered effector proteins are central to the pathogenesis of many Gram-negative bacteria, including *Salmonella enterica*, *Shigella* spp., *Yersinia pestis*, *Vibrio cholerae*, or *Pseudomonas aeruginosa* [14]. The exact role of the T3SS in infections by bordetellae, however, remains rather enigmatic. T3SS appears to play a key role in the establishment of chronic infections by *B. bronchiseptica* complex I isolates, contributing to persistent bacterial colonization of the lower respiratory tract of a wide range of mammalian hosts, including rats, mice, and pigs [15–17]. The T3SS activity of *B. bronchiseptica* complex I strains was reported to alter dendritic cell maturation, enhance the production of the anti-inflammatory cytokine interleukin-10 (IL-10), suppress antigen-specific IFN-γ production during infection and to hinder the development of anti-*Bordetella* serum antibody levels [15–19]. Yet, increased

virulence of *B. bronchiseptica* complex IV strains and their ability to cause lethal pulmonary infections in experimental animals positively correlated with enhanced expression of T3SS genes [12]. In infected macrophages and epithelial cells, the activity of T3SS of *B. bronchiseptica* accounts for rapid cell death that is non-apoptotic and is caspase-1-independent [20]. This T3SS-dependent cytotoxicity of *B. bronchiseptica* is mediated by the 69-kDa effector protein *Bb* BteA, an unusually potent cytotoxin that is translocated into target cells, localizes into ezrin-rich lipid rafts via its 130 N-terminal amino acid residues, and causes cytotoxicity via its C-terminal part by an unknown mechanism of action [21, 22].

The T3SS in *B. pertussis* species was thought to be inactive as several studies failed to demonstrate T3SS gene expression in *B. pertussis* and no T3SS-mediated cytotoxicity was usually observed *in vitro*. However, its expression was recently shown to be turned on upon passage of *B. pertussis* through mice and lost again during passaging on laboratory media [23, 24]. Indeed, low-passage human *B. pertussis* isolates appear to produce the *Bp* BteA effector [25]. Moreover, the deletion of the *btrA* regulator gene in *B. pertussis* yielded a detectable *Bp* BteA-dependent cytotoxicity towards human HeLa epithelial cells which, however, remained significantly lower than the cytotoxicity resulting from *B. bronchiseptica* infection [13].

We report here that *Bp* BteA possesses a much lower cytotoxic activity to yeast and human cells than *Bb* BteA and that this is due to the insertion of an additional alanine residue at position 503 (A503) of *Bp* BteA, which is conserved in different *B. pertussis* lineages. The *B. pertussis* mutant producing a *Bp* BteAΔA503 effector lacking A503 is shown to exhibit a strongly enhanced virulence (lower LD50) in the mouse model of intranasal infection at high inoculation doses, whereas at sublethal inoculation doses it elicited less inflammatory pathology in infected mouse lungs.

## Results

### Insertion of an alanine residue at position 503 strongly reduces the cytotoxic activity of the T3SS effector BteA

To understand the structural and functional constraints of the T3SS effector BteA, we analyzed its amino acid sequence diversity within *B. bronchiseptica* and *B. pertussis* species. This revealed a considerable extent of the *Bb* BteA sequence polymorphism in between *B. bronchiseptica* species of complex I and complex IV, as demonstrated in Fig 1A by 24 differences in *Bb* BteA sequences of RB50 complex I and D445 complex IV strains. On the contrary, protein sequences of *Bp* BteA from *B. pertussis* species were quite monomorphic. When 634 protein sequences of *Bp* BteA from the NCBI database were downloaded and analyzed, 565 of them turned out to be identical. The prototypical *Bp* BteA sequence differed only in 6 sequence positions from *Bb*RB50 BteA, as depicted in Fig 1A and was encoded for example by laboratory Tohama I strain, or recent clinical isolates B1920 and B1917, representatives of the *B. pertussis* *ptxP1-ptxA1* and *ptxP3-ptxA1* lineages, respectively.

To assess whether BteA protein sequence polymorphism translates into functional differences, we determined the ability of the three BteA variants depicted in Fig 1A to complement the cytotoxicity of the mutant Δ*bteA* *B. bronchiseptica* D445 strain. In line with previous reports, *B. bronchiseptica* D445 was highly toxic towards human HeLa epithelial cells (Fig 1B). Within 3 h of infection, 40% lysis of HeLa cells occurred at a multiplicity of infection (MOI) 5:1 whereas near-complete cell lysis resulted from infection at MOI 50:1 (Fig 1B). In contrast, no cell lysis was provoked by infection with D445-derivatives harboring in-frame deletions of the open reading frames encoding the T3SS ATPase (Δ*bscN*), or *Bb* BteA effector (Δ*bteA*), respectively. Hence, all observed cytotoxicity of the D445 strain towards HeLa cells was due to the action of the *Bb* BteA effector. As also shown in Fig 1B, *bteA* alleles of *B. bronchiseptica*

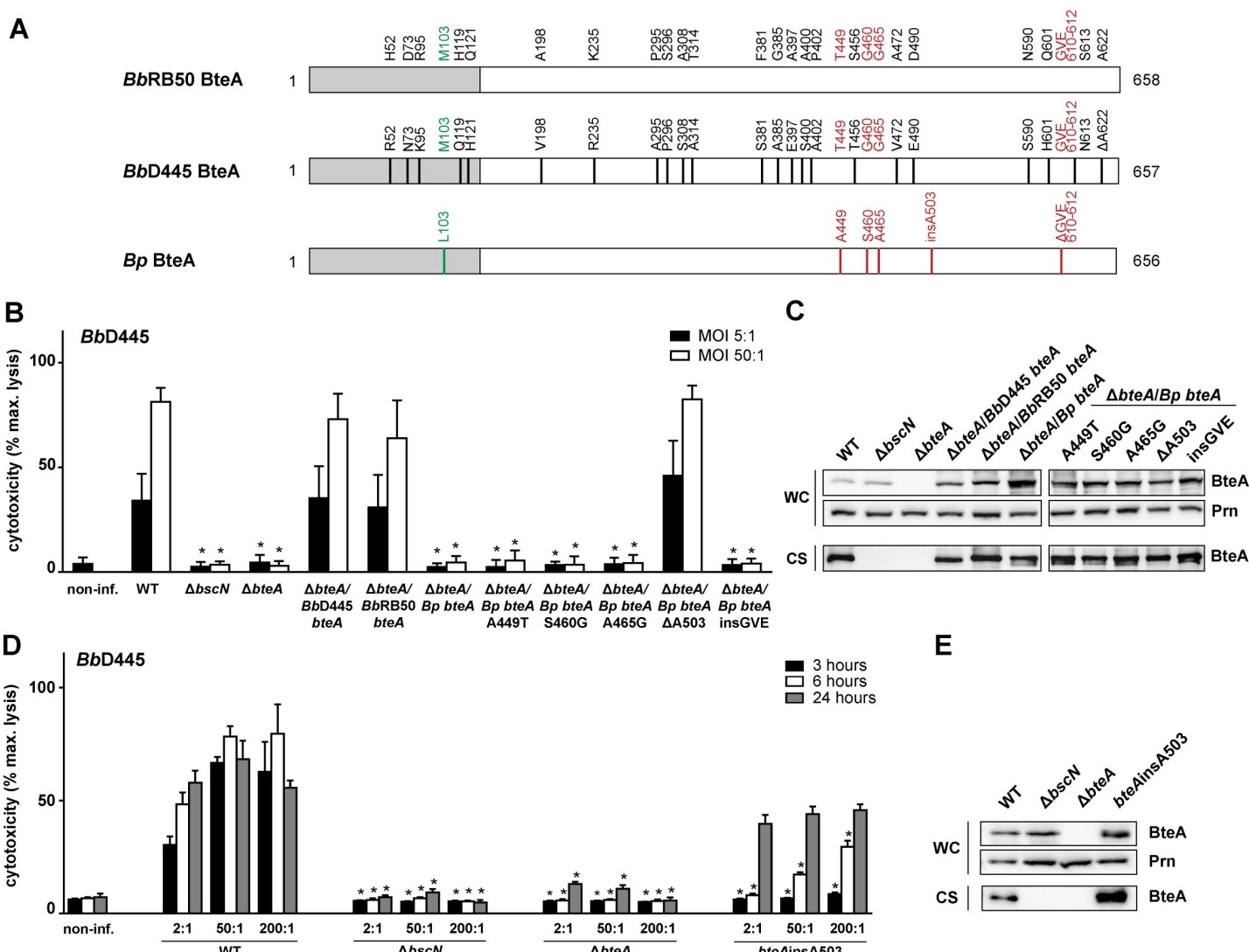

**Fig 1. The presence of Ala codon at position 503 strongly reduces the cytotoxic activity of the T3SS effector BteA.** (A) Schematic representation of BteA of *B. bronchiseptica* RB50 (*Bb*RB50 BteA, WP_003814629.1), *B. bronchiseptica* D445 (*Bb*D445 BteA, WP_004567631.1) and BteA of *B. pertussis* Tohama I, B1917 and B1920 strains (*Bp*BteA, WP_010929841.1). Differences in the primary structure are shown. Black lines represent residues uniquely present in *Bb*D445 BteA, green and red lines represent residues uniquely present in lipid raft targeting (LRT) and cytotoxic domain of *Bp*BteA, respectively. LRT domain is colored in grey and cytotoxic domain in white. (B and C) *Bp*BteA restores cytotoxicity of *bteA*-knockout *B. bronchiseptica* D445 only after deletion of Ala residue at position 503. HeLa cells were infected with the following *B. bronchiseptica* D445 strains: wild type (WT), Δ*bscN*, Δ*bteA* or Δ*bteA* strain complemented *in trans* with BbD445 *bteA* (Δ*bteA/Bb*D445 *bteA*), *Bb*RB50 *bteA* (Δ*bteA/Bb*RB50 *bteA*) and *Bp bteA* (Δ*bteA/Bp bteA*), or with mutated *Bp bteA* (Δ*bteA/Bp bteA*A449T, Δ*bteA/Bp bteA*S460G, Δ*bteA/Bp bteA*A465G, Δ*bteA/Bp bteA*ΔA503, Δ*bteA/Bp bteA*insGVE) at an multiplicity of infection (MOI) of 5:1 and 50:1. Cytotoxicity was measured as lactate dehydrogenase (LDH) release 3 h post-infection. Values represent the means ± SE for 4 independent experiments performed in duplicate ($n = 8$). $^*$ $p < 0.01$, compared with WT-infected cells, unpaired two-tailed $t$-test using the corresponding MOI. (C) Immunoblot analysis of whole-cell lysates (WC; 0.1 ml of culture equivalent; $OD_{600} = 1$) and culture supernatants (CS; 1 ml of the culture equivalent; $OD_{600} = 1$) of the indicated strains using anti-BteA (BteA, 1: 10,000) or anti-pertactin (Prn, 1:10,000) antibodies. A representative experiment out of 2 is shown. (D and E) Insertion of Ala codon into position 503 of *Bb* BteA attenuates *B. bronchiseptica* D445 cytotoxicity. HeLa cells were infected with *B. bronchiseptica* D445 WT or Δ*bscN*, Δ*bteA*, and *bteA*insA503 mutant strains at an MOI of 2:1, 50:1 and 200:1. Cytotoxicity was measured as lactate dehydrogenase (LDH) release 3, 6, and 24 h post-infection. Values represent the means ± SE for 2 independent experiments performed in duplicate ($n = 4$). $^*$ $p < 0.01$, compared with WT-infected cells, unpaired two-tailed $t$-test using the corresponding infection time and MOI. (E) Immunoblot analysis of whole-cell lysates (WC; 0.1 ml of culture equivalent; $OD_{600} = 1$) and culture supernatants (CS; 1 ml of the culture equivalent; $OD_{600} = 1$) of the indicated strains using anti-BteA (BteA, 1: 10,000) or anti-pertactin (Prn, 1:10,000) antibodies. A representative experiment out of 3 is shown.

D445 and RB50 strains expressed *in trans* from the pBBRI plasmid complemented equally well the cytotoxicity defect of the Δ*bteA B. bronchiseptica* D445 mutant. In contrast, the expression of *Bp bteA* from *B. pertussis* B1917 failed to restore the cytotoxic activity of the same Δ*bteA*

mutant. This was unexpected, as the deduced *Bp* BteA protein sequence of the B1917 strain was identical to that of the Tohama I, the *Bp* BteA of which was previously reported to be functionally interchangeable with the *Bb* BteA effector [22]. This discrepancy was not due to a problem with the production of the *Bp* BteA of the strain B1917 in our system, as the levels of the *Bp* BteA secreted by the Δ*bteA* mutant into culture supernatants were fully comparable to the levels of *Bb* BteA (Fig 1C). This suggested that by difference to what has been reported, the *Bp* BteA effector of *B. pertussis* possessed, in fact, a significantly lower specific cytotoxic activity than its *Bb* BteA counterpart from *B. bronchiseptica*.

We next reasoned that the low activity of the *Bp* BteA effector was due to one or more amino acid differences within the *Bp* BteA cytotoxic domain as compared to the cytotoxic domains of *Bb* BteA proteins of strains RB50 and D445. As highlighted in red in Fig 1A, these comprised three amino acid residue substitutions (T449A, G460S, G465A), one alanine residue insertion at position 503 (insA503) and one deletion of the GVE tripeptide at position 610 (ΔGVE 610–612). Therefore, we individually introduced three *Bb* BteA sequence-guided residue substitutions, one deletion, and one residue insertion into the *Bp* BteA sequence and expressed the five mutated *Bp* bteA alleles *in trans* in the Δ*bteA* host strain. As shown in Fig 1B, neither of the three residue substitutions (A449T, S460G, or A465G) nor insertion of the GVE tripeptide at position 610, had any impact on the very low cytotoxic activity of the *Bp* BteA protein. In contrast, deletion of the alanine residue at position 503 (ΔA503) increased the cytotoxic activity of the *Bp* BteAΔA503 protein to the level of cytotoxic activity of *Bb* BteA effectors of *B. bronchiseptica* D445 and RB50 (Fig 1B), produced and secreted at comparable levels (Fig 1C).

To corroborate the finding that insertion of the alanine residue at position 503 debilitated the cytotoxic activity of the native *Bp* BteA effector, we inserted an Ala codon into the corresponding position of the *bteA* allele on the chromosome of *B. bronchiseptica* D445. As shown in Fig 1D, an A503 insertion into *Bb* BteA attenuated *B. bronchiseptica* D445 strain cytotoxicity 3 h post-infection even at MOI 200:1 while the strain retained the same amount and integrity of the BteA protein in bacterial whole-cell lysate and culture supernatant (Fig 1E). The residual cytotoxic activity of the *Bb* BteAinsA503 protein became apparent only at 6 h post-infection with about the 50% lysis of HeLa cells occurring within 24 hours (Fig 1D). Taken together, these results suggest that the insertion of an alanine residue at position 503 strongly reduced but did not fully ablate the cytotoxic activity of the *Bb* BteA effector produced by *B. bronchiseptica*.

It was important to validate the impact of Ala 503 insertion on *Bp* BteA activity in the absence of the potentially confounding activities of the other factors of *B. bronchiseptica* and *B. pertussis*, such as that of the CyaA, DNT or PT toxins. Therefore, we constructed *Bp* BteA and *Bb* BteA protein fusions tagged at C-terminus by GFP and produced them in *Saccharomyces cerevisiae* cells, a model system sensitive to the activity of BteA [26]. The *bteA-GFP* alleles of *B. bronchiseptica* RB50 and *B. pertussis* B1917 were expressed from the galactose-inducible *GAL1* promoter in the low-copy number centromeric pYC2-CT vector. As shown in Fig 2A, irrespective of the carried *bteA-GFP* allele, all transformants of *S. cerevisiae* grew equally well on glucose-containing media, on which the *GAL1* promoter was repressed. However, only yeast producing *Bp* BteA-GFP, or the attenuated *Bb* BteAinsA503-GFP proteins, grew on agar plates containing the galactose as an inducer. The yeast producing wild type *Bb* BteA-GFP, or the *Bp* BteAΔA503-GFP fusions, failed to grow on galactose-containing plates (Fig 2A). As further shown by representative growth curves, all transformed yeast strains grew comparably well in liquid media containing glucose. However, in galactose-containing liquid media, the yeast producing *Bb* BteA-GFP or *Bp* BteAΔA503-GFP grew much slower than the yeast expressing *Bp* BteA-GFP or the *Bb* BteAinsA503-GFP variants, respectively (Fig 2B). Fluorescence microscopy examination confirmed that the *Bp* BteA-GFP and *Bb* BteAinsA503-GFP proteins

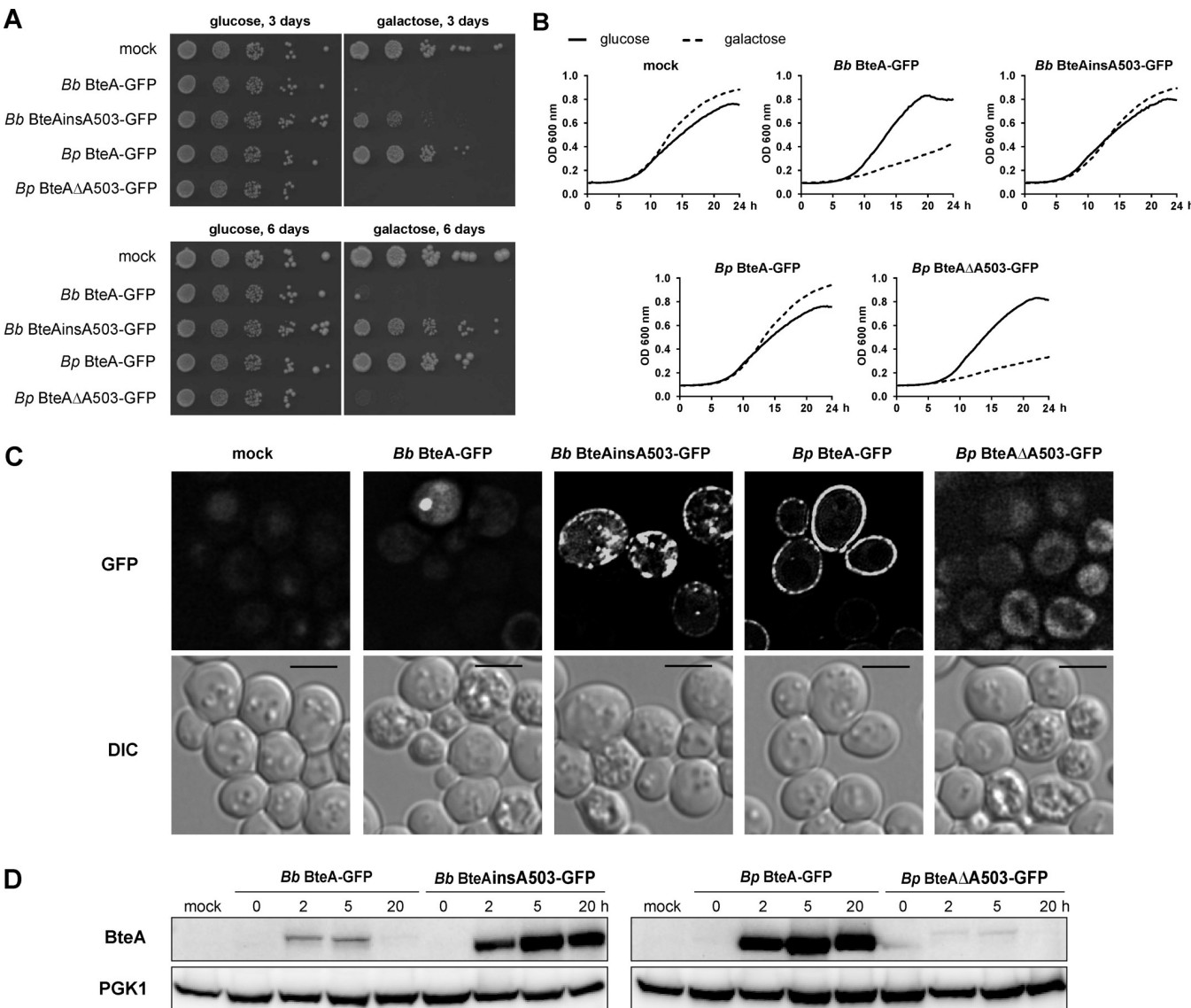

**Fig 2. BteA-induced cytotoxicity in yeast is determined by Ala codon at position 503.** (A) Spot assay of *S. cerevisiae* BY4741 strain harboring empty pYC2-CT vector (mock) or the same vector encoding indicated GFP-tagged *Bb* BteA protein variants of *B. bronchiseptica* RB50 (*Bb* BteA-GFP, *Bb*RB50 BteAinsA503-GFP), or *Bp* BteA protein variants of *B. pertussis* B1917 (*Bp* BteA-GFP, *Bp* BteAΔA503-GFP). Strains were spotted in ten-fold serial dilutions on the control glucose plates or plates with galactose to induce BteA-GFP expression. Cells were grown at 30°C for 3 and 6 days. A representative experiment out of 3 is shown. (B) Growth curves of yeast cultures with empty pYC2-CT vector (mock) or the same vector carrying indicated GFP-tagged BteA protein variants in the control glucose medium or in the medium supplemented with galactose. The optical density at 600 nm ($OD_{600}$) of cultures at 30°C was recorded every 15 min for 24 h. A representative experiment out of 2 is shown. (C) Live-cell imaging of yeast cells with empty pYC2-CT vector (mock), or the same vector carrying the indicated GFP-tagged BteA proteins variants. Yeast cells were cultivated for 20 h in the medium supplemented with galactose to induce BteA expression. Representative images from 3 independent experiments with the same outcome are presented. Scale bar, 4 μm (D) Immunoblot analysis of whole-cell lysates of yeast cells expressing indicated GFP-tagged BteA protein variants. Yeast protein extracts were prepared from non-induced culture with empty pYC2-CT vector (mock) and from induced cultures harboring indicated GFP-tagged BteA protein variants encoded on the same vector. Cultures were analyzed 0, 2, 5, and 20 h post-induction with galactose. Equal volumes of extracts (0.4 ml of the culture equivalent; $OD_{600} = 1$) were separated by SDS-PAGE and analyzed by immunoblot using anti-BteA (BteA, 1: 10,000) or anti-PGK1 (PGK1, 1: 5,000) antibodies. PGK1 was used as a loading control. A representative experiment out of 2 is shown.

localized into the plasma membrane of yeast cells and occasionally formed intracellular foci (Fig 2C). As judged from the differential interference contrast (DIC) images, yeast cells producing these proteins remained viable upon induction with galactose (Fig 2C). In contrast, a

high proportion of dead cells was observed by DIC imaging in yeast cultures induced to produce the *Bb* BteA-GFP and *Bp* BteAΔA503-GFP proteins. The fluorescence signal of these toxic proteins was hardly above the background. The *Bb* BteA-GFP and *Bp* BteAΔA503-GFP proteins were detected in yeast cells at low amounts 2 and 5 h after induction and were undetectable at 20 h post-induction, likely due to the death of producing cells (Fig 2D). In contrast, robust production of the *Bp* BteA-GFP and *Bb* BteAinsA503-GFP proteins was observed already 2 h after induction by galactose, and the proteins accumulated over 20 h of culture (Fig 2D). These data confirmed that the insertion of the A503 residue accounts for the low cytotoxic activity of *B. pertussis Bp* BteA and that elimination of A503 strongly increases the effector activity of BteA.

## Ala503 removal from *Bp* BteA enhances *B. pertussis* virulence in mouse lungs

To assess whether A503 insertion in the *Bp* BteA protein modulates the overall cytotoxicity and virulence of *B. pertussis*, we deleted the codon for the A503 residue from the *bteA* allele on *B. pertussis* B1917 chromosome. As shown in Fig 3A, at MOI 50:1 the expression of the

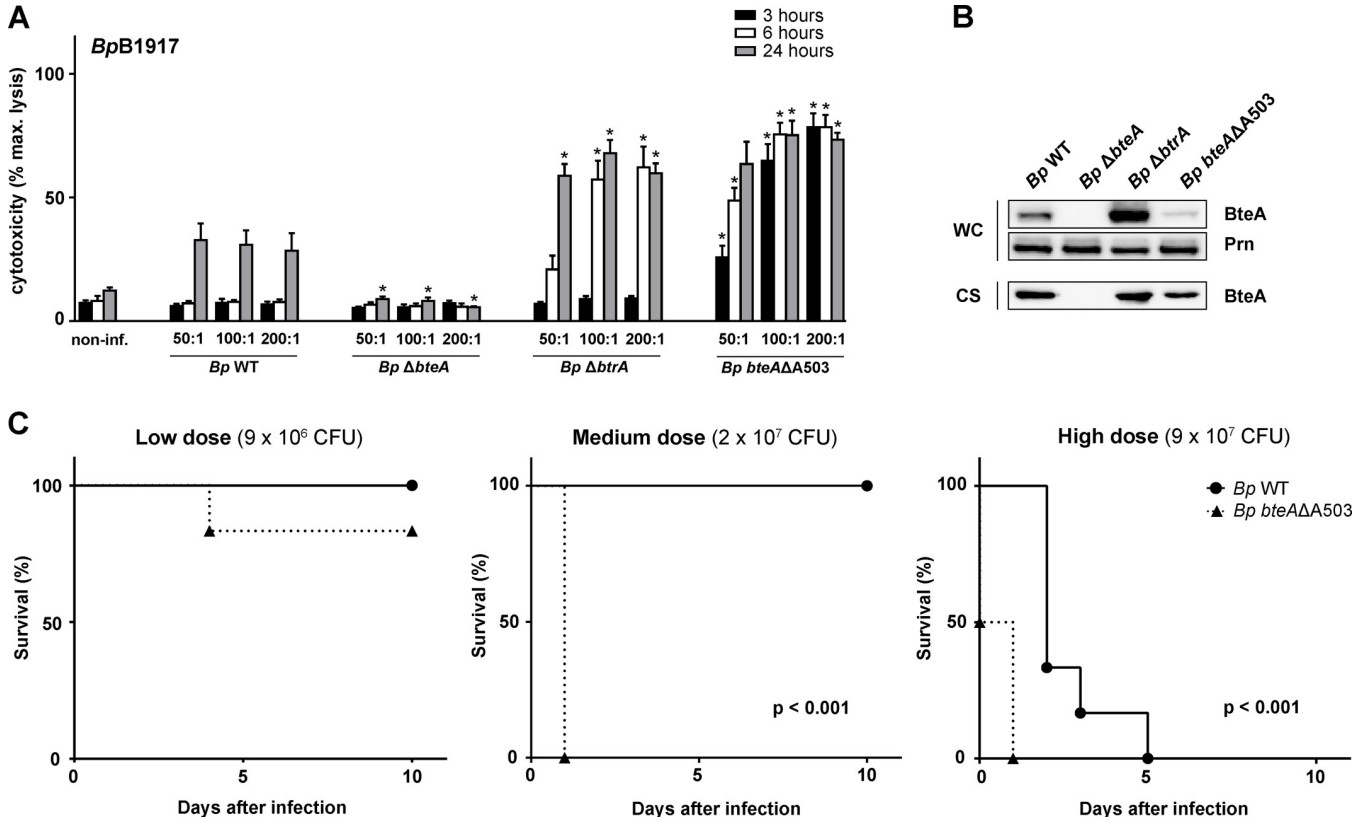

**Fig 3. Virulence of *B. pertussis* is potentiated by the removal of Ala503 residue from *Bp* BteA effector protein.** (A and B) The deletion of Ala-encoding codon 503 in the *Bp bteA* allele enhances *B. pertussis* B1917 cytotoxicity. HeLa cells were infected with *B. pertussis* B1917 wild type (*Bp* WT), Δ*bteA* (*Bp* Δ*bteA*), Δ*btrA* (*Bp* Δ*btrA*), or *bteA*ΔA503 (*Bp bteA*ΔA503) at an MOI of 50:1, 100:1 and 200:1. Cytotoxicity was measured as lactate dehydrogenase (LDH) release 3, 6, and 24 h post-infection. Values represent the means ± SE for 3 independent experiments performed in duplicate ($n = 6$). * $p < 0.01$, compared with *Bp* WT-infected cells, unpaired two-tailed *t*-test using the corresponding infection time and MOI. (B) Immunoblot analysis of whole-cell lysates (WC; 0.1 ml of culture equivalent; $OD_{600} = 1$) and culture supernatants (CS; 1 ml of the culture equivalent; $OD_{600} = 1$) of the indicated strains using anti-BteA (BteA, 1: 10,000) or anti-pertactin (Prn, 1:10,000) antibodies. A representative experiment out of 2 is shown. (C) The deletion of Ala503 residue from *Bp* BteA effector protein enhances *B. pertussis* B1917 virulence. Five-week-old female Balb/cByJ mice were intranasally infected with indicated doses of *B. pertussis* B1917 wild type (*Bp* WT) or *bteA* ΔA503 (*Bp bteA*ΔA503) derivative. Survival rates were monitored over a 10-day period. The results are representative of 3 independent experiments using 6 mice per challenged group and challenge dose. *p*-values represent significant differences in survival distributions between *Bp* WT vs. *Bp bteA* ΔA503, long-rank test using the corresponding infection dose.

*bteAΔA503* allele enabled the *B. pertussis* mutant to elicit 40% lysis of the infected HeLa cells within 3 h of infection. In contrast, a comparable extent of lysis occurred only in 24 hours following infection by the parental *B. pertussis* B1917 strain secreting native *Bp* BteA at comparable levels (Fig 3B). The *ΔbtrA* mutant with derepressed expression and secretion of native *Bp* BteA ([13] and Fig 3B) then provoked comparable cell lysis in 6 hours. These results corroborate our finding that presence of the A503 residue within the *Bp* BteA protein attenuates the overall cytotoxic activity of *B. pertussis* towards infected cells but does not fully ablate it.

To determine the impact of attenuation of BteA effector protein activity on *B. pertussis* virulence, we took advantage of the mouse model of intranasal infection, which mimics pneumonia elicited by *B. pertussis* in infants. Five-week-old female Balb/cByJ mice (6 *per* group) were intranasally infected with the wild type *B. pertussis* B1917 (*Bp* WT) and the *Bp bteAΔA503* mutant at several high inoculation doses needed for determination of the LD50 value. As documented by one representative result of the three performed challenge experiments, all mice inoculated with 9 x $10^7$ CFU of *B. pertussis* B1917 died within 5 days after infection, with the first deaths occurring on day 3 post-infection. At the lower challenge doses of 2 x $10^7$ and 9 x $10^6$ CFU all mice infected with wild type *B. pertussis* B1917 survived till day 10 (Fig 3C). This yielded a calculated LD50 value of (4.4 ± 0.72) x $10^7$ of *B. pertussis* B1917 CFU, in good agreement with the LD50 value reported for infections of Swiss CD-1 mice with the *B. pertussis* Tohama I strain [27]. In contrast, all mice inoculated with 2 x $10^7$ or 9 x $10^7$ CFU of the *Bp bteAΔA503* mutant succumbed to the infection already by day 1 and the first deaths occurred already within hours from infection. The challenge with the *Bp bteAΔA503* mutant also repeatedly caused death of some of the infected mice even at the reduced dose of 9 x $10^6$ CFU (Fig 3C). The calculated LD50 value of (1.3 ± 0.21) x $10^7$ CFU for the *Bp bteAΔA503* mutant is significantly lower than the LD50 value of (4.4 ± 0.72) x $10^7$ for the wild type strain (n = 3, p < 0.01, unpaired two-tailed *t*-test). Nevertheless, only 3-fold difference of this value likely does not fully reflect the real increase of virulence of the mutant, as the *Bp bteAΔA503*-infected mice presented obvious early symptoms of severe illness, such as ruffled fur and lethargy, and died from infection much earlier than mice infected with the wild type *B. pertussis* B1917 (Fig 3C). Accordingly, long-rank test for survival distribution at high and medium inoculation dose revealed significant differences (p < 0.001, *Bp* WT vs. *Bp bteA* ΔA503).

## *B. pertussis* producing the more active BteAΔA503 effector triggers lower inflammatory damage of infected lungs

To determine the role of A503 insertion in the establishment of *B. pertussis* infection, we next assessed the lung colonization capacity of the *Bp bteAΔA503* mutant at a sublethal challenge dose and performed the histological examination of the infected lung tissue. As shown in Fig 4A, after intranasal inoculation of 1 x $10^5$ CFU/mouse, the *Bp bteAΔA503* mutant reproducibly colonized mouse lungs to the same level as the wild type *B. pertussis* B1917 (*Bp* WT). Both strains reached $10^6$ CFU *per* infected lung on day 6 or 7 post-infection with the subsequent clearing of the bacterial load below $10^3$ CFU on day 21. The more cytotoxic *bteAΔA503* mutant thus did not exhibit a higher capacity to proliferate in mouse lungs than the parental strain.

The histological examination of lung tissues was performed at the peak of infection at day 6. To allow for rigorous sampling of the lung pathology, lungs were fixed *in situ* and approximately 12 transverse lung sections for each mouse were made over the entire organ. As exemplified in representative micrographs of the hematoxylin-eosin (HE)-stained specimens in Fig 4B, in comparison to non-infected lungs (control) exhibiting the typical structure of lung parenchyma with alveolar septa, patent alveoli and well-preserved bronchi and bronchioles, infection with the wild type *B. pertussis* B1917 strain elicited bronchopneumonic lesions.

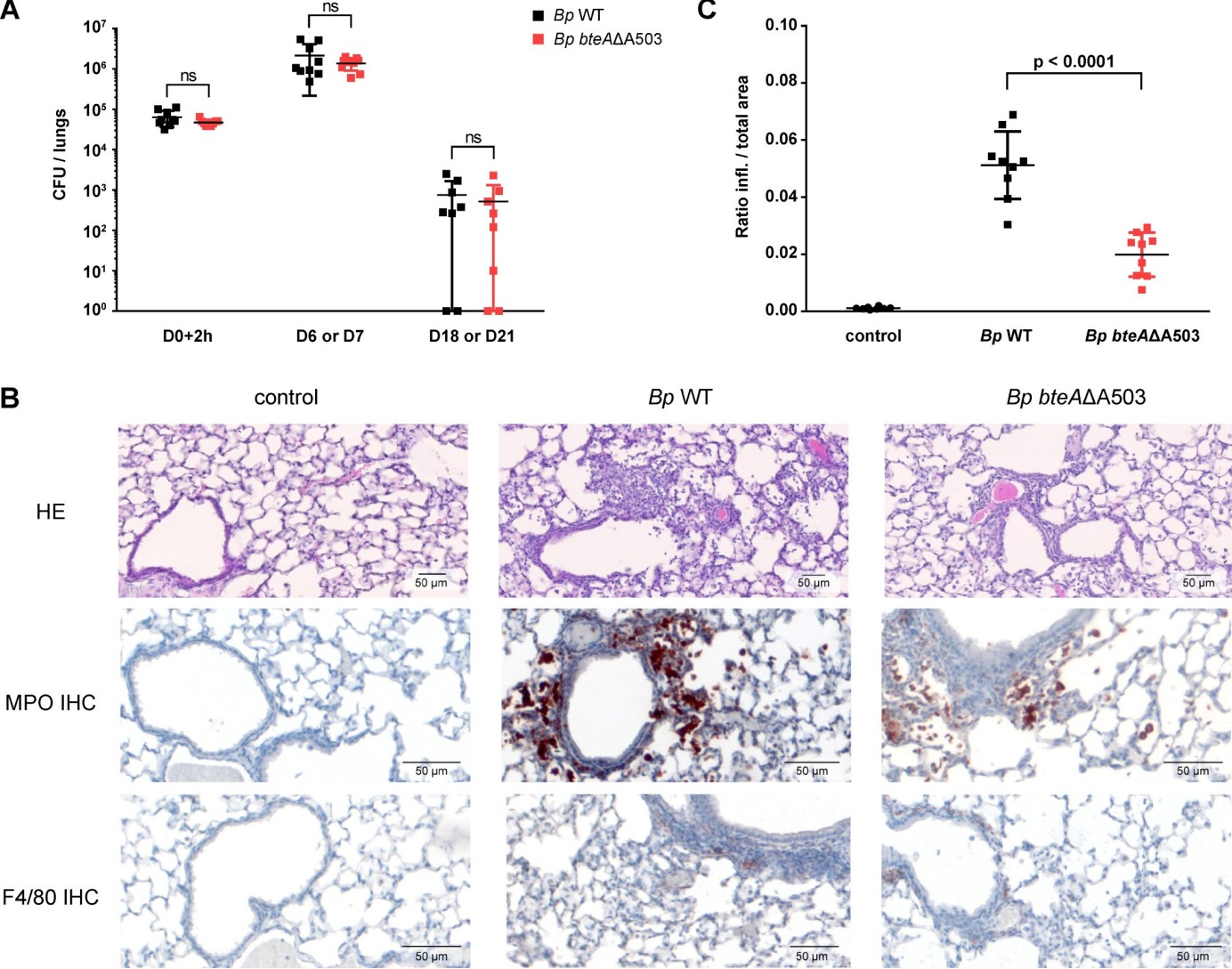

**Fig 4. Deletion of Ala503 residue from *Bp* BteA effector protein limits *B. pertussis*-induced bronchopneumonia.** (A) The *B. pertussis bteAΔA503* derivative strain colonizes mice lungs as efficiently as wild type strain. Five-week-old female Balb/cByJ mice (3 to 4 mice per time point and group) were intranasally infected with $10^5$ CFU of *B. pertussis* B1917 wild type (*Bp* WT) or *bteAΔA503* (*Bp bteAΔA503*), and bacterial load in their lungs was determined. Data are pooled from 3 independent experiments, and points represent CFUs counts of individual lungs at indicated days (D) after the exposure to challenge suspension. The determination was performed in exp.1 on D0+2h, D6, D18, in exp.2 on D0+2h, D6, D21, and in exp.3 on D0+2h and D7. The means ± SD are shown (n = 8 to 9 per time point and group). ns, not significant, *Bp* WT vs. *Bp bteA* ΔA503, unpaired two-tailed *t*-test. (B-C) Deletion of Ala503 residue from *Bp* BteA effector protein limits *B. pertussis* B1917-induced bronchopneumonia. Mice infected intranasally with $10^5$ CFUs of *Bp* WT or *Bp bteAΔA503* were sacrificed on day 6, and their lung tissue was examined upon hematoxylin and eosin (HE) staining. The presence of neutrophils and macrophages was revealed by myeloperoxidase (MPO) and F4/80 immunohistochemical (IHC) staining, respectively. (B) Representative images of lung parenchyma. (C) Regions of the lung parenchyma containing inflammatory infiltrate were manually delimited in a blinded manner on 12 consecutive HE-stained lung sections for each animal, and the ratio of inflamed (infl.) to total parenchyma areas was calculated for each lung. Data shown are from 2 independent experiments, each containing 5 mice per group (n = 10). The mean value ± SD is shown. p < 0.0001, *Bp* WT vs. *Bp bteA* ΔA503, unpaired two-tailed *t*-test.

These presented as the accumulation of inflammatory cells inside alveoli and around bronchi and bronchioles, and contained polymorphonuclear (neutrophilic) granulocytes and numbers of monocytes/macrophages as revealed upon immunohistochemical staining for myeloperoxidase (MPO) of neutrophils and the F4/80 marker of macrophages (Fig 4B). As also shown in Fig 4B by representative micrographs, the sublethal infection by the *Bp bteAΔA503* mutant elicited a strikingly milder lung pathology with less pronounced infiltration of neutrophils and

macrophages than was observed in samples of lungs infected by the parental strain. The Qupath software was used to obtain a quantitative measurement of the severity of the induced bronchopneumonia [28]. The inflamed regions of HE-stained specimens in all lung sections from 2 independent experiments on total of 10 animals per group were manually delimited in a blinded manner. The ratio of inflamed parenchyma area to total parenchyma was next calculated for each lung specimen and was averaged per animal group. This yielded the mean portion of the inflamed area of 5.97% for the *B. pertussis* B1917-infected lungs. A significantly lower ratio of 1.47% ($p$<0.0001) was obtained for mouse lungs infected with the *bteAΔA503* mutant strain (Fig 4C). Hence, the production of the more active *Bp* BteAΔA503 T3SS effector conferred on *B. pertussis* B1917 an enhanced capacity to attenuate induced inflammatory pathology and damage of mouse lungs.

## Discussion

We report here that the T3SS-injected effector *Bp* BteA of the human pathogen *B. pertussis* exhibits a much lower *in vitro* cytotoxic activity and *in vivo* capacity to attenuate bronchopneumonia than its *Bb* BteA counterpart from the animal pathogen *B. bronchiseptica*. The reduction of *Bp* BteA activity could be attributed to the insertion of an additional alanine residue at position 503 of *Bp* BteA sequence. Our data suggest that *B. pertussis* and *B. bronchiseptica* BteA primary structure divergence might lead to different immune-modulating properties of their BteA *in vivo*, possibly contributing to the diverging nature of the respiratory infections caused by these species.

The pathogenic *Bordetella* species appear to have evolved from an environmental ancestor upon acquisition of virulence factors that enabled specific interactions with animal hosts [29–31]. One such event was the acquisition of loci encoding the type III secretion system (T3SS) which allows for the delivery of bacterial effector proteins directly into the cytosol of host cells. T3SS is the most conserved secretion system of classical *Bordetella* species as determined by the genome analysis of 11 sequenced classical *Bordetella* strains [32]. The vast majority of nucleotide substitutions observed between *B. pertussis* and *B. bronchiseptica* T3SS loci are silent or result in conservative amino acid substitutions. The high degree of conservation of the T3SS structural genes suggests a strong purifying selection on the T3SS-encoding locus and an important role of T3SS in the lifestyle of these *Bordetella* species [32, 33].

Interestingly, the T3SS-mediated cytotoxicity of *B. pertussis* to infected epithelial cells *in vitro* is usually negligible, when compared to the cytotoxicity caused by *B. bronchiseptica* infection. Previously, this phenotype was attributed to the differences in the function of the BtrA-BtrS regulatory node that controls the T3SS gene expression [13]. Indeed, in-frame deletion of the anti-σ factor *btrA* gene was shown to liberate a negative control over the T3SS gene expression in *B. pertussis* and induce its T3SS-dependent cytotoxicity ([13] and Fig 3A). Moreover, French and coworkers previously reported that the alleles encoding the BteA effectors responsible for *Bordetella* T3SS-induced cytotoxicity are functionally interchangeable in between *B. bronchiseptica* and *B. pertussis* species [22]. However, such conclusion is at odds with our herein presented results. We used two independent approaches to demonstrate that because of an additional alanine residue inserted at position 503, the *Bp* BteA protein of *B. pertussis* possesses a much lower specific cytotoxic activity than the *Bb* BteA protein of *B. bronchiseptica*. To show this, we expressed the *bteA* alleles of both species in the same *ΔbteA* strain of *B. bronchiseptica* and compared their cytotoxic activities. We further demonstrated that removal of the A503 residue from the poorly active *Bp* BteA protein enhances its cytotoxic activity and *vice versa*, that insertion of A503 into the cytotoxic *Bb* BteA effector decreases its activity. This occurred without effects on BteA expression levels or the efficacy of its T3SS-

mediated secretion. We further corroborated these results by determination of the cytotoxicity of the heterologously produced intact *Bb* BteA and *Bp* BteA proteins, and their respective A503-inserted/deleted variants, in yeast cells. We are therefore confident that the specific cytotoxic activities of *Bp* BteA and *Bb* BteA proteins differ substantially due to the acquisition of the A503 residue in *Bp* BteA. Moreover, as demonstrated in Fig 3A by comparison with the Δ*btrA* mutant of *B. pertussis* B1917, the insertion of A503 into *Bp* BteA plays a major role in the attenuation of the *B. pertussis* overall cytotoxic activity towards infected cells.

Remarkably, when we analyzed 634 protein sequences of *Bp* BteA present in the NCBI database, we found that 565 of them are identical to the sequence of *Bp* BteA from *B. pertussis* B1917. Strikingly, the additional alanine 503 within *Bp* BteA protein is also present in the *Bp* BteA sequences that are different from *Bp* BteA of the strain B1917, e.g. *Bp* BteA of the strain 18323. On the contrary, as shown in S1 Fig, all distinct subpopulations of *B. bronchiseptica* complex I and complex IV, and all diverse ovine and human *B. parapertussis* isolates lack this additional alanine at position 503 in their *Bb* BteA and *Bpp* BteA proteins, respectively. Interestingly, the primary structure of the classical *Bordetella* species comprises additional alanine at position 502 (S1 Fig), suggesting that the acquisition of alanine 503 in *Bp* BteA of *B. pertussis* species could occur via duplication of alanine 502. This duplication event very likely occurred early in the *B. pertussis* specialization to the human population, therefore being conserved within *B. pertussis* complex II lineage I (*B. pertussis* 18323) and *B. pertussis* complex II lineage II (*B. pertussis* Tohama I, *B. pertussis* B1917, *B. pertussis* B1920) but absent in human *B. parapertussis* complex III and *B. bronchiseptica* complex IV. However, the BteA structure and the mechanisms behind its cytotoxic action remain unknown and how the insertion of A503 into *Bp* BteA reduces its specific cytotoxicity remains to be determined.

Further work is also needed to answer the intriguing questions on the biological relevance of A503 insertion into the *Bp* BteA of *B. pertussis* and on the role of T3SS in the pathogenesis of pertussis as such. It is assumed that *B. pertussis* evolved from a *B. bronchiseptica*-like ancestor to become a strictly human-adapted pathogen, causing an acute illness [10]. In contrast, *B. bronchiseptica* bacteria have a broader host range and typically cause rather chronic and often asymptomatic respiratory tract infections in a wide variety of mammals [3, 9]. The T3SS activity of *B. bronchiseptica* was shown to be involved in persistent colonization of the trachea and lungs in mouse, rat, and pig models of infection [15–17]. The underlying mechanism probably involves the induction of IL-10-producing cells early in the infection and modulation of dendritic cell maturation [18, 19]. This might subsequently alter adaptive immune responses required for the clearance of *B. bronchiseptica*. Indeed, the T3SS activity of *B. bronchiseptica* inhibits the generation of IFNγ-producing cells and hinders the development of anti-*Bordetella* serum antibodies [16, 19]. Further, T3SS-mediated suppression of the NF-*κB* pathway activation might downregulate the expression of innate immune genes encoding anti-microbial peptides, and also proinflammatory cytokines and chemokines, thereby attenuating recruitment of inflammatory cells to the sites of *Bordetella* infection [16, 34, 35]. In line with that, we show here that the *bteA*ΔA503 mutant of *B. pertussis*, producing a *Bp* BteAΔA503 protein with strongly increased cytotoxic activity, elicited strikingly less bronchopneumonic lesions in the infected mouse lungs than the same bacterial load of the parental *B. pertussis* B1917 strain (*c.f.* Fig 4A and 4C). Hence, the presence of an additional alanine residue at position 503 in *Bp* BteA reduces the capacity of the *Bp* BteA effector to inhibit activation of host pro-inflammatory responses in the *B. pertussis*-infected lungs. It remains to be determined how this would affect the course of *B. pertussis* infection in humans, namely the capacity of *B. pertussis* to colonize the mucosa of upper airways and to induce nasopharyngeal catarrh accompanied by sneezing and coughing that facilitates *B. pertussis* spread in the human population. These crucial pathophysiological manifestations of human pertussis are not reproduced in mice.

In the sublethal mouse lung infection model used here, the strong enhancement of the *Bp* BteA activity following the deletion of the A503 residue did not translate into any enhanced proliferation, or an extended persistence of the *Bp bteAΔA503* mutant, which was cleared at a similar rate as the parental *B. pertussis* bacteria (*c.f.* Fig 4A). It thus remains possible that the redundant immunosuppressive activities of the other potent toxins produced by *B. pertussis*, such as the adenylate cyclase (CyaA) or pertussis (PT) toxins, may have masked the contribution of the increased activity of *Bp* BteAΔA503 to immune evasion and persistence of the *bteAΔA503* mutant in infected lungs. However, when high numbers of bacteria in the range of the LD50 dose were used to infect mice, the *bteAΔA503* mutant exhibited an increased virulence. The mutant strain provoked a more severe course of the disease, with accelerated death of animals challenged with lower infectious doses than those of the wild type strain. This goes well with the observation that the activity of T3SS and delivery of the *Bb* BteA effector into host cells plays an important role in hypervirulence of *B. bronchiseptica* strains [11, 12].

In conclusion, our data show that the more active *Bp* BteAΔA503 effector was able to importantly intervene in the interactions of *B. pertussis* bacteria with host defense, and its action shaped the course and outcome of the infection. It is tempting to speculate that acquisition of the A503 residue and thus resulting attenuation of the *Bp* BteA activity, represents an evolutionary adaptation that fine-tunes pathogenesis of *B. pertussis* to induce the whooping cough disease in humans.

## Materials and methods

### Bacterial strains and growth conditions

The bacterial strains used in this study are listed in S1 Table. The *Escherichia coli* strain XL1-Blue was used throughout this work for plasmid construction and the *Escherichia coli* strain SM10 λ pir was used for plasmid transfer into bordetellae by bacterial conjugation. *E. coli* strains were cultivated at 37˚C on Luria-Bertani (LB) agar medium or in LB broth. When appropriate, LB culture media were supplemented with ampicillin (pSS4245 and pYC2-CT plasmid transformants, 100 μg/ml) or chloramphenicol (pBBRI plasmid transformants, 30 μg/ml for *E. coli* XL-1 Blue transformants, 15 μg/ml for *E. coli* SM10 λ pir transformants). The *B. bronchiseptica* D445 was generously provided by Dimitri Diavatopoulos (Radboud Center for Infectious Diseases, Nijmegen, The Netherlands), *B. pertussis* B1917 was a kind gift of Frits R. Mooi (National Institute of Public Health and the Environment, Bilthoven, The Netherlands) and *B. bronchiseptica* RB50 was generously provided by Branislav Vecerek (Institute of Microbiology, Prague, Czech Republic). *Bordetella* strains were grown on Bordet-Gengou (BG) agar medium (Difco, USA) supplemented with 1% glycerol and 15% defibrinated sheep blood (LabMediaServis, Jaromer, Czech Republic) at 37˚C and 5% $CO_2$ for 48 h (*B. bronchiseptica*) or 72 h (*B. pertussis*), or in modified Stainer-Scholte (SSM) medium [36] supplemented with 5 g/l of Casamino Acids at 37˚C. To maximize the expression of the type III secretion system for assays, SSM medium was formulated with reduced L—glutamate (monosodium salt) concentration (11.5 mM, 2.14 g/l) and without the addition of $FeSO_4.7H_2O$ [37, 38]. Bacteria for assays and inoculations were grown to a mid-exponential phase (*B. bronchiseptica* $OD_{600}$ 1.5, *B. pertussis* $OD_{600}$ 1.0) at 37˚C. Samples for protein analysis in bacterial pellets and culture supernatants were taken at the late exponential / early stationary phase (*B. bronchiseptica* $OD_{600}$ 3.0–4.0, *B. pertussis* $OD_{600}$ 1.5–2.0). When appropriate, the culture medium was supplemented with 30 μg/ml of chloramphenicol to maintain the pBBRI *bteA* plasmid.

### Plasmid construction

Plasmids used in this study are listed in S2 Table and were constructed using T4 DNA ligase or the Gibson assembly strategy [39]. PCR amplifications were performed from chromosomal

DNA of *B. pertussis* B1917, *B. bronchiseptica* RB50, and *B. bronchiseptica* D445, respectively using Herculase II Phusion DNA polymerase (Agilent, USA). The site-directed substitutions, insertions, and deletions were introduced into the *bteA* gene by PCR mutagenesis. All constructs were verified by DNA sequencing (Eurofins Genomics, Germany).

### *Bordetella* allelic exchange and allele complementation

The suicide allelic exchange vector pSS4245, which contains an *StrR* allele functional in bordetellae but not in *E. coli*, was used for an allelic exchange on the *Bordetella* chromosome, as described in detail previously [27]. The first crossing-over into the *Bordetella* chromosome was selected on BG agar supplemented with 50 mM MgSO$_4$, 500 µg/ml of streptomycin and 150 µg/ml of ampicillin, and 100 µg/ml of kanamycin (*B. bronchiseptica*) or 30 µg/ml of ampicillin, and 40 µg/ml of kanamycin (*B. pertussis*). To select for the second crossing-over *Bordetella*, colonies were re-streaked on BG agar lacking MgSO$_4$ and antibiotics. The presence of introduced mutations was verified by PCR amplification of relevant portions of *Bordetella* chromosome followed by agarose gel analysis and DNA sequencing (Eurofins Genomics, Germany). PCR primers used for verification of introduced mutations are listed in S3 Table. The pBBRI plasmid constructs expressing different *bteA* alleles under the control of *B. bronchiseptica* or *B. pertussis bteA* promotor were used for *bteA* complementation of the mutant Δ*bteA B. bronchiseptica* D445 strain. Complemented *B. bronchiseptica* strains were selected on BG agar medium supplemented with 60 µg/ml of chloramphenicol and 10 µg/ml of cephalexin to which bordetellae are naturally resistant and re-streaked on the BG agar medium supplemented with 30 µg/ml of chloramphenicol to maintain the pBBRI *bteA* plasmid.

### Protein analysis in bacterial whole-cell lysates and culture supernatants

*Bordetella* strains were cultivated in the SS medium at 37˚C. For analysis of whole-cell lysates, bacterial culture aliquots were centrifuged (30 min; 30,000 g) and pellets were lysed in 8 M urea and 50 mM Tris-HCl, pH 8.0. Bacterial DNA and cellular debris were removed by centrifugation and extracted proteins were mixed with SDS-PAGE sample loading buffer. To analyze supernatant fractions, aliquots of cell-free bacterial culture supernatant were precipitated with 10% trichloroacetic acid overnight at 4˚C, washed with acetone, dissolved in 8 M urea, 50 mM Tris-HCl, pH 8.0 and mixed with SDS-PAGE sample loading buffer. Heat-boiling of samples before SDS-PAGE electrophoresis was avoided to prevent irreversible aggregation of BteA protein and high molecular weight smears. Samples with ODs equivalent to 0.1 OD$_{600}$ unit (whole-cell lysates) or 1 OD$_{600}$ unit (bacterial supernatants) were separated by SDS-PAGE electrophoresis (10% gel) and transferred onto a nitrocellulose membrane. Membranes were probed overnight with mouse polyclonal antibodies raised against BteA (dilution 1:10,000) or rabbit polyclonal antibodies raised against pertactin (dilution 1:10,000, both kindly provided by Branislav Vecerek, Institute of Microbiology, Prague, Czech Republic). The detected proteins were revealed with 1:3,000-diluted horseradish peroxidase (HRP)-conjugated anti-rabbit or anti-mouse IgG secondary antibodies (GE Healthcare) using a Pierce ECL chemiluminescence substrate (Thermo Fisher Scientific, USA) and an Image Quant LAS 4000 station (GE Healthcare, USA).

### Mammalian cell culture and cytotoxicity assay

HeLa cells (ATCC CCL-2, human cervical adenocarcinoma) were grown in Dulbecco's Modified Eagle Medium (DMEM) supplemented with 10% (vol/vol) heat-inactivated fetal bovine serum (FBS) at 37˚C and 5% CO$_2$. For cytotoxicity assay, 1 x 10$^5$ HeLa cells per well were seeded into 96-well plate in DMEM supplemented with 2% (vol/vol) FBS without phenol red

indicator and allowed to attach overnight. *Bordetella* culture grown to a mid-exponential phase was added at the indicated multiplicity of infection (MOI) and centrifuged onto HeLa cell monolayers (5 min, 400 g). Following the incubation at 37°C and 5% $CO_2$ for the indicated times, cytotoxicity of *Bordetella* infection was measured as lactate dehydrogenase (LDH) release into cell culture media using CytoTox 96 assay (Promega) according to the manufacturer's instructions. The % cytotoxicity was calculated using the following equation: ($OD_{495}$ sample–$OD_{495}$ media)/($OD_{495}$ total lysis–$OD_{495}$ media)* 100.

## Yeast strains, growth conditions, and spot assay

*S. cerevisiae* BY4741 (*MAT*a; *his3*Δ1; *leu2*Δ0; *ura3*Δ0; *met15*Δ0) strain (Euroscarf, Germany) was used in this study for heterologous galactose-inducible expression of BteA-GFP fusion proteins. The wild type and derivative *bteA* alleles were cloned downstream from the *GAL1* promoter in frame with the C-terminal GFP gene in a low copy centromeric vector pYC2-CT (Invitrogen, USA) carrying the *URA3* selection marker. The generated *S. cerevisiae* strains are listed in S4 Table, whereas pYC2-CT-derived plasmids are listed in S2 Table. Yeast cell transformation was carried out according to the one-step protocol of Chen *et al*. 1992 [40]. Yeast cells containing pYC2-CT derivatives were grown at 30°C on solid minimal SC agar or in liquid SC media containing appropriate dropout mix and 2% glucose (SD-Ura media, non-inducing media) or 1% raffinose + 2% galactose (SC-Ura-galactose media, inducing media). For spot assays, ten-fold serial dilutions of *S. cerevisiae* cultures from a starting density of $10^7$ cells/ml were performed using SC-Ura-galactose media, and cell dilutions were then spotted by a 48-pin tool on SD-Ura or SC-Ura-galactose agar plates. Plates were incubated at 30°C and photographed after 3 and 6 days. To analyze the growth of yeast cells in liquid cultures, yeast cells were inoculated from liquid SD-Ura overnight cultures into 1.5 ml of SD-Ura or inducing SC-Ura-galactose media to $OD_{600}$ = 0.05–0.1 and cultivated with agitation at 30°C in a plate reader (BioTek EON) using flat-bottom 12-well plates (Greiner). The culture density $OD_{600}$ was recorded every 15 min.

## Fluorescence microscopy

To reduce the background fluorescence during live-cell imaging, yeast cells were washed in low-fluorescence-synthetic complete medium (LF-SC) and mounted on a microscope cover glass by covering with a thin slice of 1.5% agarose prepared in LF-SC. Widefield microscopy was performed using an Olympus IX-81 inverted microscope equipped with a motorized stage, a 100x PlanApochromat oil-immersion objective (NA = 1.4), and a Hamamatsu Orca-ER-1394 digital camera. GFP fluorescence was detected with the filter block U-MGFPHQ, exc. 460–488 nm, em. 495–540 nm. Z-stacks were taken with 0.28 μm z-steps and deconvolution was performed with Advanced Maximum Likelihood (AMLE) filters (Xcellence Imaging SW, Olympus). Images were collected in a 16-bit format, processed using Olympus Cell-R Xcellence, and assembled with help of the Adobe CS5 software package.

## Protein analysis in yeast cells

Yeast cells harboring pYC2-CT derivatives encoding BteA-GFP fusion proteins were cultivated in SC-Ura-galactose media for 2, 5, and 20 h to induce the BteA expression. Equivalents of $OD_{600}$ = 1 of yeast cell cultures were collected and denatured yeast protein extracts were prepared by NaOH lysis/TCA precipitation method according to Volland *et al. 1994* [41]. Equal amounts of protein extracts were separated by SDS-PAGE electrophoresis (10% gel) followed by the transfer onto a nitrocellulose membrane. Membranes were probed overnight with mouse polyclonal antibodies raised against BteA (dilution 1:10,000, kindly provided by

Branislav Vecerek, Institute of Microbiology, Prague, Czech Republic) or mouse monoclonal antibody against yeast phosphoglycerate kinase PGK1 (anti-PGK1, dilution 1:5,000, Abcam, ab113687) as a loading control. The detected proteins were revealed with 1:3,000-diluted horseradish peroxidase (HRP)-conjugated anti-mouse IgG secondary antibody (GE Healthcare) using a Pierce ECL chemiluminescence substrate (Thermo Fisher Scientific, USA) and an Image Quant LAS 4000 station (GE Healthcare, USA).

## Animal infection experiments

Five-week-old female Balb/cByJ mice (Charles River, France) were used in this study. Mice were anesthetized by intraperitoneal (i.p.) injection of ketamine (80 mg/kg) and xylazine (8 mg/kg) and intranasally infected with the indicated colony forming units (CFUs) of mid-exponential phase *B. pertussis* B1917 or its *bteA*ΔA503 derivative delivered in 50 μl volume. To determine viable CFUs of the *B. pertussis* inoculum, aliquots of the inoculum were diluted in PBS and plated on BG agar plates.

For survival curves, groups of six Balb/cByJ mice were infected with $9 \times 10^6$, $2 \times 10^7$ or $9 \times 10^7$ CFUs / mouse and mice survival was monitored over a 10-day period. The results are expressed as the percentage of surviving mice and are representative of 3 independent experiments. To determine the significance of the differences between survival distributions longrank test was used, and *p* values were generated using GraphPad Prism 6. The LD50 values were calculated by the probit analysis method of Finney, as previously reported [27].

To determine *B. pertussis* lung colonization capacity, Balb/cByJ mice were infected with $1 \times 10^5$ CFUs / mouse. Mice were then euthanized 2 h (D0 + 2 h) and on indicated days after exposure to challenge suspension. Lungs were excised and homogenized in sterile PBS using tissue grinders. Dilutions of lung homogenates were plated on BG agar plates. CFUs were counted after 5 days of incubation at 37˚C and 5% $CO_2$. The data come from 3 independent experiments, where three to four mice per time point per group were used.

## Histopathological analysis

Lung morphology was examined 6 days upon intranasal challenge of Balb/cByJ mice with $10^5$ CFU of *B. pertussis* B1917 or its *bteA*ΔA503 derivative. Fixation and preparation of paraffin sections of the lungs were carried out as previously described [27]. In brief, mice were anesthetized by i. p. injection of ketamine (80 mg/kg) and xylazine (8 mg/kg) and sacrificed by cervical dislocation. An incision was made along the left parasternal line to open the chest, the trachea was cannulated with an 18-Gauge cannula and the lungs were instilled with 4% (wt/vol) buffered formaldehyde. After removal from the cadaver, the lungs were immersed into 4% (wt/vol) buffered formaldehyde solution for 24 h, followed by transfer into 70% (vol/vol) ethanol solution for storage. Specimens were embedded into 6% buffered agarose gel cubes and sliced using a PLA 3D printed matrix (DeltiQ L, TRILAB, Czech Republic) as described by Tyson *et al.* 2015 [42]. Approximately twelve 2 mm thick transversal lung slices were prepared from the entire organ. While keeping the orientation succession the slices were further processed into paraffin blocks using the Leica ASP6025 tissue processor and Leica EG1150H+C tissue embedding centre (both Leica Biosystems, Nussloch, Germany). Three serial adjacent 2.5 μm thick sections were then cut with the Leica RM 2255 Microtome (Leica Biosystems, Nussloch, Germany) and mounted on standard microscopy slides (VWR International) or SuperFrost Plus slides (Thermo Scientific, Waltham, MA, USA). Slides were stained with Hematoxylin and eosin (HE) (Sigma-Aldrich, MO, USA) in the automated Leica ST5020+CV5030 Multistainer/Coversliper (Leica Biosystems, Nussloch, Germany). For the immunohistochemical (IHC) staining, the histological specimens were rehydrated and retrieved in the pH 6 HIER

buffer (Zytomed GmbH, Berlin, Germany), followed by staining with rabbit anti-myeloperoxidase (MPO) monoclonal antibody (dilution 1:1,000, clone CI:EPR20257, Abcam, United Kingdom) or rat anti-F4/80 monoclonal antibody (dilution 1:200, clone Cl:A3-1, Abcam, United Kingdom). The detection of antibody binding was performed with anti-rabbit IgG HRP-conjugated polymer (Zytomed GmbH, Berlin, Germany) or anti-rat IgG HRP-conjugated polymer ImmPRESS (Vector Laboratories, Burlingame, CA, USA) and 3-amino-9-ethylcarbazole (AEC) solution (Zytomed GmbH, Berlin, Germany. IHC-stained slides were counterstained with Hematoxylin solution (Sigma-Aldrich, MO, USA). Stained samples were mounted using Aquatex mounting medium (Merck-Millipore, Darmstadt, Germany). Slides were scanned using the AxioScan.Z1 automated slide scanner (Carl Zeiss, Göttingen, Germany) and representative images were generated using QuPath software [28]. Quantification of pulmonary lesions was performed in a blinded manner. Total area and regions containing inflammatory infiltrate in HE-stained specimens were manually delimited using QuPath software. The relative proportion of the inflamed area to the total area was calculated. The data come from 2 independent experiments, each containing 5 animals per group.

### Statistical analysis

The significance of the differences between groups was determined by unpaired two-tailed *t*-test. Exact group size is indicated in the figure legends (n), and *p* values were generated using MATLAB R2019b. Differences were considered statistically significant at $p < 0.01$.

### Ethics statement

The European Community Council Directive 86/609/EEC, Appendix A of the Council of Europe Convention ETS123, and the Czech Republic Act for Experimental Work with Animals (Decree No. 207/2004 Sb, and the Acts Nos. 246/92 Sb and 77/2004 Sb) were obeyed during the laboratory animal care and experiments. Permission no. 38/2018 was issued by the Animal Welfare Committee of the Institute of Molecular Genetics of the Czech Academy of Sciences in Prague.

### Supporting information

**S1 Fig. All isolates of *B. pertussis* carry alanine at position 503 in their *Bp* BteA protein unlike isolates of *B. bronchiseptica* and *B. parapertussis* species.** Alignment of amino acid residues 488–519 of BteA effector protein of representative *Bordetella* species. *Bp* BteA of *B. pertussis* complex II lineage I: 18323 (*Bp*18323, WP_014905434.1), complex II lineage II: Tohama I (*Bp*Tohama I, WP_010929841.1), B1917 (*Bp*B1917, WP_010929841.1), B1920 (*Bp*B1920, WP_010929841.1), *Bb* BteA of *B. bronchiseptica* complex I: RB50 (*Bb*RB50, WP_003814629.1), 253 (*Bb*253, WP_003814629.1), complex IV: D445 (*Bb*D445, WP_004567631.1), Bbr77 (*Bb*Bbr77, WP_003820735.1), and *Bpp* BteA of *B. parapertussis* ovine: Bpp5 (*Bpp*Bpp5, WP_003814629.1) and human: 12822 (*Bpp*12822, WP_010929242.1) is shown. Presence of alanine at position 503 is highlighted in red.
(PDF)

**S1 Table. List of bacterial strains used in this study.** Bacterial strain name, genotype description and reference are indicated.
(PDF)

**S2 Table. List of plasmids used in this study.** Plasmid name, description and reference are provided.
(PDF)

**S3 Table. List of PCR primers used for verification of *Bordetella* chromosome mutations.**
The analyzed strain, primer sequence (5′-3′) and its position in chromosome are provided.
(PDF)

**S4 Table. List of mammalian and yeast cells used in this study.** Name, description and reference are indicated.
(PDF)

## Acknowledgments

We thank H. Lukeova, I. Marsikova, and L. Novakova for their excellent technical help.

## Author Contributions

**Conceptualization:** Jana Kamanova.

**Data curation:** Jan Bayram, Ivana Malcova, Jana Kamanova.

**Formal analysis:** Jan Bayram, Ivana Malcova, Jana Kamanova.

**Funding acquisition:** Radislav Sedlacek, Peter Sebo, Jana Kamanova.

**Investigation:** Jan Bayram, Ivana Malcova, Larisa Sinkovec, Jana Holubova, Gaia Streparola, David Jurnecka, Jana Kamanova.

**Methodology:** Jan Bayram, Ivana Malcova, Jana Holubova, Gaia Streparola, Jan Kucera, Jana Kamanova.

**Project administration:** Jana Kamanova.

**Resources:** Ivana Malcova, Radislav Sedlacek, Peter Sebo.

**Software:** Jan Bayram.

**Supervision:** Jana Kamanova.

**Validation:** Jan Bayram, Ivana Malcova, Jana Kamanova.

**Visualization:** Jan Bayram, Ivana Malcova, Jana Kamanova.

**Writing – original draft:** Jana Kamanova.

**Writing – review & editing:** Peter Sebo, Jana Kamanova.

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
