## [Decision Letter · Decision Letter 0]

22 May 2020

Dear Dr. Kamanova,

Thank you very much for submitting your manuscript "Cytotoxicity of the effector protein BteA was attenuated in Bordetella pertussis by insertion of an alanine residue" for consideration at PLOS Pathogens. Your manuscript was reviewed by members of the editorial board and by two independent reviewers. The reviewers appreciated the attention to an important topic. Based on the reviews, we are likely to accept this manuscript for publication, providing that you modify the manuscript according to the review recommendations. These recommendation are considered minor and can be done without the need for additional experimentation. 

Sincerely,

Brian K Coombes

Associate Editor

PLOS Pathogens

Renée Tsolis

Section Editor

PLOS Pathogens

Kasturi Haldar

Editor-in-Chief

PLOS Pathogens

orcid.org/0000-0001-5065-158X

Michael Malim

Editor-in-Chief

PLOS Pathogens

orcid.org/0000-0002-7699-2064

Reviewer Comments (if any, and for reference):

Reviewer's Responses to Questions

**Part I - Summary**

Reviewer #1: The paper by Bayram et al., deals with the cytotoxicity of BteA, a T3SS

effector. The authors show that a single alanine makes a great

differences between to bordetella strains, B. pertussis and B.

bronchiseptica, in which this addition lower the effector cytotoxicity.

Reviewer #2: Discovered more than a decade ago, the Bordetella BteA effector and the T3SS that delivers it have remained enigmatic. On the one hand they are conserved between and especially within B. pertussis and B. bronchiseptica lineages, yet BteA/T3SS functionality in B. pertussis seems almost cryptic when compared to B. bronchiseptica isolates. This raises the questions of why it's conserved and what might it really be doing? Although regulation is likely a factor, this study clarifies and extends the story in an important way. The data clearly show that a single amino acid insertion in BteA (Ala503), which is nearly universal in B. pertusis lineages, attenuates its cytotoxic activity in vitro and alters the course of virulence in mice. Perhaps most striking, however, is that the converse is also shown to be true - simply eliminating this residue increases the cytotoxic activity of BteA by nearly two orders of magnitude. B. pertussis is an evolving pathogen and we currently use suboptimal vaccines, understanding how mutations affect virulence, function, and selection is key for surveillance and for devising better vaccines.

**Part II – Major Issues: Key Experiments Required for Acceptance**

Reviewer #1: The paper is clearly written and has all the needed controls that I

could think of. Although the finding are important to give an idea for

the cause of differences between the activity of the two homologs, the

authors lack a good explanation for this and do not give a mechanism for

this changes. I do miss some secondary structure analysis that may

indicate for the position of this insertion and if it is exposed or

located in a place that might do something.

Reviewer #2: In general, the experiments are carefully performed, the data are convincing, and the manuscript is well written. I recommend, however, that the authors address the following suggestions:

1. Although the A503 insertion is highly attenuating, residual activity can be detected as shown in Fig. S1. The authors should elevate this histogram to become Fig. 1E. It shows a comparatively low level of cytotoxic activity for Bp BteA when the T3SS is derepressed. The residual cytotoxic activity could simply be that, or it could be due to some other activity of BteA that only indirectly affects viability. Regardless, the function, if any, of the most commonly circulating B. pertussis BteA allele seems important to thoroughly understand.

2. The Discussion could be tightened up in a few places and would benefit from some pruning. In addition, clearly pointing out the discrepancy between the data presented here and the French et al. 2009 paper is important, but I suggest toning it down a bit. There are many factors that might account for such differences, and the data presented in this manuscript are clear and convincing.

3. The survival plots in Fig. 3C and 3D need appropriate statistics. All I could find is a general statement that the results were reproduced in 2 experiments using 6 mice per challenged group and challenge dose.

**Part III – Minor Issues: Editorial and Data Presentation Modifications**

Reviewer #1: (No Response)

Reviewer #2: (No Response)

PLOS authors have the option to publish the peer review history of their article (what does this mean?). If published, this will include your full peer review and any attached files.

Reviewer #1: No

Reviewer #2: No
---

## [Editor Report · Decision Letter 1]

24 Jun 2020

Dear Dr. Kamanova,

We are pleased to inform you that your manuscript 'Cytotoxicity of the effector protein BteA was attenuated in Bordetella pertussis by insertion of an alanine residue' has been provisionally accepted for publication in PLOS Pathogens.

Best regards,

Brian K Coombes

Section Editor

PLOS Pathogens

Renée Tsolis

Section Editor

PLOS Pathogens

Kasturi Haldar

Editor-in-Chief

PLOS Pathogens

orcid.org/0000-0001-5065-158X

Michael Malim

Editor-in-Chief

PLOS Pathogens

orcid.org/0000-0002-7699-2064
---

## [Editor Report · Acceptance letter]

29 Jul 2020

Dear Dr. Kamanova,

We are delighted to inform you that your manuscript, "Cytotoxicity of the effector protein BteA was attenuated in *Bordetella pertussis* by insertion of an alanine residue," has been formally accepted for publication in PLOS Pathogens.

Best regards,

Kasturi Haldar

Editor-in-Chief

PLOS Pathogens

orcid.org/0000-0001-5065-158X

Michael Malim

Editor-in-Chief

PLOS Pathogens

orcid.org/0000-0002-7699-2064